# Evaluating Seasonal Rainfall Forecast Gridded Models over Sub-Saharan Africa

**Winifred Ayinpogbilla Atiah** [1,2,*] **, Eduardo Garcia Bendito** [3] **and Francis Kamau Muthoni** [3]

1 Meteorology Department (MISU), Stockholm University, 106 91 Stockholm, Sweden
2 Department of Meteorology and Climate Science, Kwame Nkrumah University of Science and Technology (KNUST), Kumasi AK-039-5028, Ghana
3 International Institute of Tropical Agriculture (IITA), Nairobi P.O. Box 30709-00100, Kenya; e.bendito@cgiar.org (E.G.B.); f.muthoni@cgiar.org (F.K.M.)
* Correspondence: winifred.atiah@misu.su.se

## Abstract

Changes in the amount and distribution of rainfall highly impact agricultural production in predominantly rainfed farming systems in Africa. Reliable rainfall forecasts on a daily timescale are vital for in-season decision-making. This study evaluated the relative prediction abilities of the European Centre for Medium-Range Weather Forecasts Season 5.1 (ECMWFSv5.1) and the Climate Forecast System version 2 (CFSv2) gridded rainfall models across Africa and three sub-regions from 2012–2022. The results indicate that the performance of both models declines with increasing lead times and improves with aggregated or coarser temporal resolutions. ECMWFv5.1 consistently represented observed daily rainfall better than CFSv2 at all lead times, particularly in West Africa. On dekadal timescales, ECMWFv5.1 outperformed CFSv2 across all sub-regions. CFSv2 tended to overestimate low- and high-intensity rainfall events, whereas ECMWFv5.1 slightly underestimated low-intensity rainfall but accurately captured high-intensity events. While ECMWFv5.1 showed superior skill overall, model reliability was generally limited to West Africa; in contrast, both models performed poorly in East Africa. The high probability of detection (POD) indicates that the models are generally effective at identifying rainy days. However, their overall accuracy in forecasting rainfall across Africa varies depending on lead time, region, rainfall intensity, and elevation. While we did not apply bias-correction methods in this study, we recommend that such techniques be used in future work to improve the reliability of forecasts for operational and sectoral applications. This study therefore highlights both the strengths and the limitations of CFSv2 and ECMWFv5.1 for climate impact assessments, particularly in West Africa and low-elevation regions.

**Keywords:** validation; rainfall forecast models; CFSv2; ECMWFv5.1; Africa

## 1. Introduction

Climatic change and variability are anticipated to increase in Africa due to prevalent global warming [1,2]. This will almost certainly negatively influence agricultural production by reducing yields and altering the suitability of crop and livestock species [3]. Rainfall is a critical driver of many socio-economic activities, particularly in African countries [4,5]. This is because rain-fed agriculture is their primary source of income. It is, nonetheless, extremely subject to climate change and unpredictability. Africa is particularly vulnerable to these effects and suffers the most due to its over-reliance on rainfall for agricultural

activity. The shifts in the onset, cessation, and duration of the rainy season significantly influence farming calendar activities such as planting dates [6,7].

Reliable daily precipitation forecasts for Africa could be immediately valuable in several industries [8]. For instance, long-range accurate forecasts would greatly aid farmers, policy-makers, and the management of agricultural value chains. Given that a sizable amount of the region depends on rain-fed agriculture, such forecasts could help farmers decide on timely plowing, planting, harvesting (typically small-scale) irrigation, and animal management [9]. The livelihoods of a sizeable population segment are at risk due to unpredictability in weather forecasts, particularly for rain, which can cause misalignment of the timing of cropping calendar activities and missed opportunities to maximize crop yields. Precise knowledge of the start of the rainy season could reduce the additional costs of re-sowing seeds due to the season's false commencement [10]. Accurate forecasts also aid in planning fertilizer application, irrigation, and spraying with agrochemicals. The information is particularly relevant for policy-making and early warning systems to assess the required relief funds in case of rainfall extremes like droughts or floods. Drought conditions are more likely if the rainy season arrives ten days late [11]. Early drought detection and warning can aid planning efforts to save lives and livelihoods. More accurate rainfall data would help manage water resources, produce power, prevent disease and floods, and enhance road safety.

Despite this vulnerability, there still needs to be more accurate rainfall forecasts in Africa, where rain-fed agriculture is the dominant paradigm. This is because there are few rain gauges in this area, characterized by irregular distribution, short-term records, and substantial data gaps [12,13]. Rainfall forecasts from satellites and models are increasingly being used instead of or in addition to gauge data. Rainfall forecasts from satellites and models supplement the sparse gauge networks to assist farmers in making decisions that would boost crop yields and, hence, food security [14–17]. Rainfall projections from models are quite dense and can be used to identify regions where agricultural productivity is at risk of climate change and fluctuation. It is critical to determine areas or locations most vulnerable to changes in seasonal calendars caused by climate change and variability to help guide the evidence-based targeting of suitable climate-smart technology [6].

Seasonal predictions are crucial for managing water resources, whereas medium-range (1–15 days) forecasts are crucial for operational choices like reservoir management. Recently, the availability and applicability of medium-range climate models have increased. Examples of such models include, but are not limited to, the European Centre for Medium-Range Weather Forecast Season 5 (ECMWF5; [18]), the Global Forecast System (GFS; [19], Climate Forecast System version 2 (CFSv2; [20]), Global Ensemble Forecast System (GEFS; [21], and the Global Spectral Model (GSM; [22]). These models are associated with some uncertainties; thus, it is necessary to determine their performance before application, which is typically done by comparing the forecasts to the actual precipitation [4,23,24].

Several studies have evaluated the performance of seasonal forecasting over various geographical locations [25]. However, for the CFSv2 and ECMWF5 climate models, there have been limited evaluation studies over the African continent. Samala et al. [26] evaluated the skill of 1-month lead time forecasts of weak Indian monsoons based on the CFSv2 model. The results showed the poor skill of the CFSv2 model in capturing the observed rainfall and circulation anomalies during weak monsoons. Lorenz et al. [27] assessed the CFSv2's capacity to forecast rapid drought intensification by comparing the baseline skill acquired using only recent observations to the skill obtained by adding CFSv2 anticipate. The results demonstrated that incorporating the CFSv2 model output resulted in only a 14% increase in variance explained across the central and eastern United States. The slight gain in skill was attributable to inadequate ability in the CFSv2 forecasts rather than a

time lag in the U.S. Drought Monitor (USDM) response to ground conditions. Over South America, ref. [28] examined the ability of the ECMWF5 seasonal precipitation predictions over South America and discovered that the ECMWF5 climate forecasts are potentially useful and should be considered when planning various strategic activities. An analysis of ECMWF5 seasonal climate forecasts for Australia using a novel forecast calibration method suggests that the monthly forecasts with zero lead-time (forecasts at the start of a month for that month) can be helpful in Australian applications. Nonetheless, the performance levels vary across Australia, with high-skill areas changing with the seasons [29]. Despite the importance of this rainfall forecast to Africa's socioeconomic progress, few studies on their competency have been conducted. As a result, this research aims to assess the performance of the CFSv2 and ECMWF5 models over Africa and its three sub-regions (West, East, and Southern Africa).

## 2. Materials and Methods

### 2.1. Study Area

This validation study was carried out across Africa and its three sub-regions (Figure 1). These sub-regions are broadly defined as East Africa (9° S–25° N, 20°–50° E), Southern Africa (9°–35° S, 20°–55.5° E), and West Africa (20° W–20° E, 18°–20° S). This sub-regional demarcation was required since the rainfall regimes in these three zones vary and are influenced by complex topography, mesoscale processes, and other factors. West Africa is presented with a rectangular red box with the label A. Rainfall dynamics in this region are variable across the three climatic zones. Sub-humid weather prevails along the coast of Guinea with 1250 to 1500 mm of rain precipitation per year as the average in this area [30]. A semiarid region, the Savannah zone, receives 750–1250 mm of rainfall each year [30]. The Sahel region's brief rainy season between June and September is distinctive. With an annual rainfall average of around 750 mm, this season is unimodal [31]. Eastern Africa is labeled B on Figure 1. The area's topography is the most complicated on the continent [32]. Due to its diverse geography and significant inland water bodies, East Africa has highly complex seasonal rainfall patterns [33]. The movement of southwesterly monsoon winds and the Somali jet, which impact rainfall in this region, have been proven to be influenced by topography. On the other hand, Lake Victoria generates mesoscale circulation patterns that result in nighttime and afternoon rainfall regimes in the lake's western and eastern halves, respectively [34]. Rainfall is impacted by the intertropical convergence zone's seasonal north–south movement (ITCZ). Due to the ITCZ's migration, the area has four distinct rainy seasons: December–February, March–May, June–September, and October–December [32]. A total number of 88 stations with rainfall data spread across the Eastern Africa region were used in this study. The West African and Indian monsoons have a small area of influence in Eastern Africa. The West African monsoon is tangentially related to the summer rainfall pattern, notably over Ethiopia and South Sudan. From November to March, the Indian monsoon produces low-level easterly/northeasterly flow. From May to October, it primarily has a southerly flow (southwesterly in Northern Hemisphere and southeasterly in Southern Hemisphere). However, in the impacted regions, the peak monsoon months (such as July to September and December to February) coincide with the dry seasons [34]. The southern part of Africa is labeled C and spans the subtropics at the confluence of tropical, subtropical, and temperate weather systems by its location. As a result, it has different year-round, seasonal, and summer rainfall zones [35]. A total number of 48 stations with rainfall data from region C was used in this study.

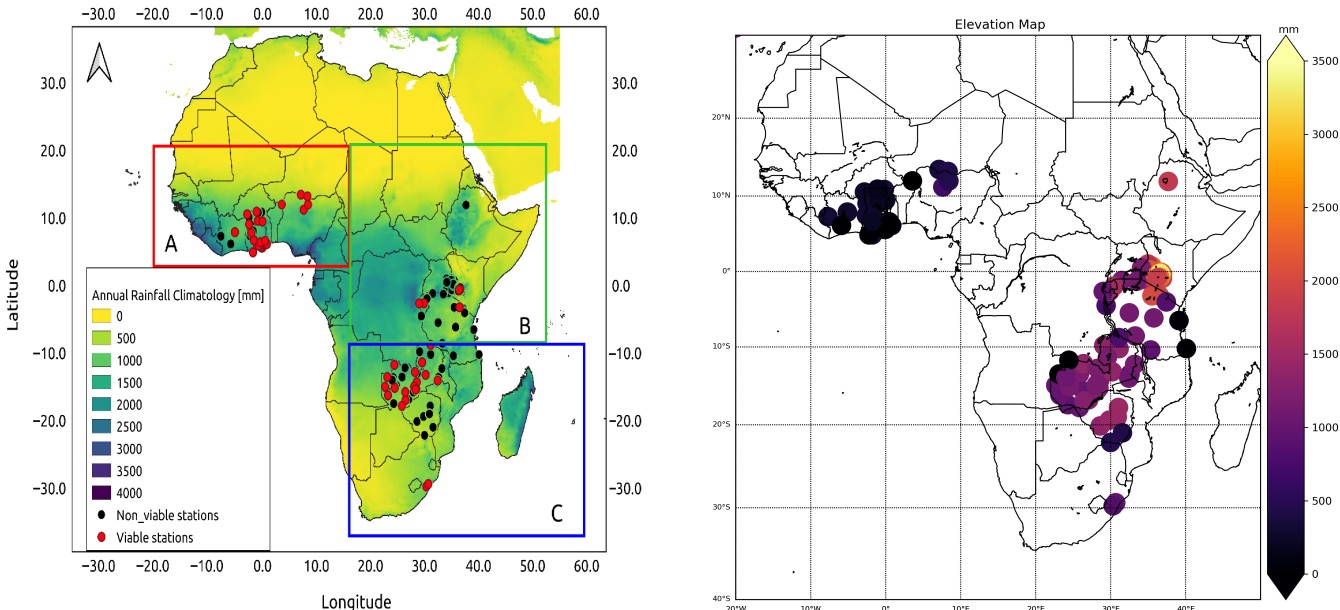

**Figure 1.** Map showing the location of observation stations (black points) overlaid on total annual rainfall from CHIRPS averaged over 1981–2020 (**left**). The three boxes show the outline of the three sub-regions, West Africa (A), East Africa (B), and Southern Africa (C), respectively. Map showing the elevations of the various gauge stations over Africa (**right**).

*2.2. Data Source*

2.2.1. Gauge Data

From 2012 through to 2022, daily rainfall fields were collected from a number of meteorological stations located in three regions in Africa (Figure 1). The gauge data was sourced from national meteorological agencies and private entities. The initial 176 stations were gathered in Africa; however, only 52 stations (29 in West Africa, 19 in South Africa, and 4 in East Africa; Table 1) were selected for the analysis after data quality control and cleaning. Specifically, stations with less than 60% of available observations over the period 2012–2022 were excluded. This threshold was chosen to maintain a balance between spatial coverage and reliability, ensuring that model evaluation is based on sufficiently complete observational records. The datasets used throughout the three sub-regions assumed a Gamma distribution.

**Table 1.** Total number of gauge observations by elevation levels, showing station counts and regional distribution.

| | Elevation (m) | | | |
|---|---|---|---|---|
| | **L1 ($<$500)** | **L2 (500–1000)** | **L3 (1000–1500)** | **L4 ($>$1500)** |
| Region | West Africa | East Africa | South Africa | South Africa |
| No. of Stations | 28 | 5 | 15 | 4 |

2.2.2. Model Data

The Climate Forecast System (CFSv1) was established by the National Centers for Environmental Prediction (NCEP). CFSv1 was a fully coupled ocean–land–atmosphere dynamical seasonal forecast system that began operations in August 2004 [36]. Its atmospheric component was a lower-resolution variant of the GFS, which was NCEP's operational global weather prediction model in 2003, while the ocean component used the Geophysical Fluid Dynamics Laboratory (GFDL) Modular Ocean Model version 3 (MOM3). In March 2011, the

system was upgraded to CFSv2, which improved nearly all aspects of data assimilation and model components [20]. Compared to its predecessor, CFSv2 demonstrated improved skill in forecasting global sea surface temperatures (SSTs) and extended the skillful prediction of the Madden–Julian Oscillation (MJO) from 6 to 17 days. It also provided a broader range of products for sub-seasonal and seasonal forecasting as well as retrospective predictions for calibration. The CFSv2 is run multiple times daily using the latest atmospheric and oceanic observations from satellites, radiosondes, ships, buoys, and other platforms [37]. By solving the physical equations governing the atmosphere, oceans, and land, it simulates weather and climate evolution over time. This system supports meteorologists and researchers in weather forecasting, climate prediction, and environmental modeling [36], and aids decision-making across sectors including agriculture, energy, and transportation.

The ECMWF5 began operation in November 2017, replacing the ECMWF-S4 [38,39], which had been in service since 2011. ECMWF5 is a substantially upgraded prediction system compared to its predecessor, including enhanced atmospheric and ocean models at higher resolution [38]. Since ECMWF-S4, the Integrated Forecast System (IFS) atmospheric model has improved, particularly in representing tropical convection [40]. Forecasts are generated for seven months using a 51-member ensemble initialized on the first of each month [41]. This version incorporates IFS Cycle 43r1, which includes improvements to the ocean model, atmospheric resolution, and land-surface initialization. The atmospheric component has a horizontal and vertical resolution of TCo319L91 (36 km grid spacing), while the ocean component uses the Nucleus for European Modelling of the Ocean (NEMO v3.4.1) [41,42].

In November 2022, ECMWF5 was upgraded to ECMWF5.1 after interpolation to a new 1-degree horizontal resolution grid that aligned the product with most of the existing forecast models [43]. Therefore our analysis utilizes the ECMWF5.1 data obtained from the Copernicus Climate Change Service Climate Data Store (C3S; [43]).

A summary of the CFSv2 and ECMWFv5.1 precipitation datasets utilized, including their main characteristics, is provided in Table 2.

**Table 2.** Comparison of CFSv2 and ECMWFv5.1 datasets, highlighting key characteristics relevant to precipitation and climate studies.

| Feature | ECMWFv5.1 | CFSv2 |
| --- | --- | --- |
| Forecast type | Seasonal forecast | Medium-range to sub-seasonal |
| Temporal resolution | Daily | Sub-daily |
| Time coverage | 1993–present | 2012–present |
| Ensemble members | 51 | 40 |
| Lead time | 215 days | 182 days |
| Grid resolution (lat $\times$ lon) | $1° \times 1°$ | $0.937° \times 0.947°$ |

Data sources: ECMWFv5.1—Copernicus Climate Data Store https://cds.climate.copernicus.eu/datasets/seasonal-original-single-levels?tab=download accessed on 3 September 2025. CFSv2—NOAA Climate Forecast System https://www.ncei.noaa.gov/data/climate-forecast-system/access/operational-9-month-forecast/6-hourly-flux/ accessed on 3 September 2025.

### 2.3. Validation Methodology

This section describes the approach used to assess the performance of the two rainfall forecast model over Africa and the three sub-regions.

The validation was carried out based on four main strategies: (a) lead-time-based assessments, (b) temporal scale assessments (daily and dekadal), (c) sub-regional assessments (West Africa, East Africa, and Southern Africa), and (d) altitudinal gradient analysis. Three lead times were considered. The first month's lead time (LT0) corresponds to the first day of the forecast to 30 days. The two-month lead time (LT1) ranges from day 31

to 60, and the three-month lead time (LT2) covers day 61–90 from the initialization of the forecasts. Rainfall forecasts for different lead times were matched with observed rainfall at the gauge stations.

Model values were collocated to gauge locations by extracting the value of the model grid cell overlapping each gauge. A model grid cell represents an areal average while a gauge is a point measurement; thus, the point-to-pixel approach may introduce representativeness bias [44]. The ideal solution is to interpolate the gauge data into grids with the resolution matching the model data. However, accurate interpolation requires a high density of quality-controlled gauge stations along the topographical and elevational gradients [45]. Given the low density of reliable gauge stations at our disposal, we opted to use the point-to-pixel method, although this could potentially introduce systematic biases in the evaluation.

For the temporal scale analysis, daily model and gauge observation data were aggregated into 10-day means to facilitate dekadal assessments of model performance. A digital elevation model (DEM) representing height above sea level was used to classify the study area into four elevation zones: <500 m (L1), 500–1000 m (L2), 1000–1500 m (L3), and >1500 m (L4). These thresholds were selected to capture the distinct orographic effects on precipitation-forming processes and patterns through their influence on thermal regimes and atmospheric circulation. The DEM was obtained from the Shuttle Radar Topography Mission (SRTM) with a spatial resolution of 30 m (1 arc-second). Table 1 shows the total gauge observations for the respective elevation levels.

### 2.4. Validation Statistics

The accuracy of the models against gauge data was evaluated using a variety of volumetric and categorical metrics.

The modified Kling–Gupta Efficiency (KGE′) was employed because it provides a single aggregated performance measure while separately quantifying correlation, bias, and variability errors. Unlike standard metrics such as RMSE, which primarily measure overall error magnitude, KGE′ allows a more nuanced evaluation of model performance, making it suitable for both deterministic and ensemble-based forecasts. The KGE′ statistic gives an aggregated score and is further decomposed into three scores that include correlation (r), bias ($\beta$), and variability ratio ($\gamma$).

1. The Pearson product–moment correlation coefficient (r). This is a statistical measure of the strength of a linear relationship between the observation and the simulations. It ranges from $-1$ to 1, but the ideal value is r = 1. The r value assesses the temporal agreement between the model and the gauge.
2. The beta ($\beta$) is the ratio between the simulated values' mean and the observed mean. A $\beta$ value greater than 1 indicates over-estimation bias, while lower than 1 indicates under-estimation compared to the gauge observations.
3. The variability ratio ($\gamma$) is computed using the coefficient of variation (variance) of forecast models and observations. The ideal value for KGE and its three indices is 1.

$$r = \frac{\sum_{i=1}^{n}(x_i - \bar{x})(y_i - \bar{y})}{\sqrt{\sum_{i=1}^{n}(x_i - \bar{x})^2}\sqrt{\sum_{i=1}^{n}(y_i - \bar{y})^2}} \tag{1}$$

$$KGE' = 1 - \sqrt{(r-1)^2 + (\beta-1)^2 + (\gamma-1)^2} \tag{2}$$

$$\beta = \frac{\mu_s}{\mu_0} \tag{3}$$

$$\gamma = \frac{CV_s}{CV_o} = \frac{\sigma_s/\mu_s}{\sigma_0/\mu_o} \qquad (4)$$

where $KGE'$ is the modified KGE-statistic (dimensionless), $r$ is the correlation coefficient between simulated and observed results (dimensionless), $\beta$ is the bias ratio (dimensionless), $\gamma$ is the variability ratio (dimensionless), $\mu$ is the mean rainfall, CV is the coefficient of variation (dimensionless), $\sigma$ is the standard deviation of rainfall, and the indices $s$ and $o$ represent simulated and observed rainfall values, respectively.

The $KGE'$ is increasingly applied for rainfall evaluation [24,28,46] since the decomposition of the model error into these three components enables a more detailed understanding of model behavior beyond a single overall error metric. For instance, a model can show good correlation if it sufficiently captures the temporal patterns of rainfall, yet it may still have a high bias if it consistently overestimates or underestimates rainfall amounts. In contrast, the standard RMSE provides only a single value that blends all error types, making it harder to pinpoint the specific aspect in which a model underperforms.

Root Mean Square Error (RMSE) measures the average magnitude of prediction errors, giving more weight to larger errors. Bias indicates the average tendency of a model to over- or underpredict, while Multiplicative Bias reflects the ratio between total predicted and observed values. A significance test was performed to assess whether the statistical metrics differed meaningfully, using a 99% confidence level with a significance threshold of $p < 0.01$.

Probability of Detection (POD) measures the fraction of observed events correctly forecasted; ideal value: 1 (perfect detection) [47]. False Alarm Ratio (FAR) indicates the fraction of predicted events that did not occur; ideal value: 0 (no false alarms). Critical Success Index (CSI) measures the proportion of observed and/or forecasted events that were correctly predicted. The Frequency Bias Index (FBI) indicates whether a model tends to overpredict ($>1$) or underpredict ($<1$) events, with an ideal value of 1. Table 3 shows the definition and calculation of statistical metrics employed.

**Table 3.** Description of other volumetric and categorical evaluation statistics used in this study. The terms A, B, and C represent hits, false alarms, and misses, respectively; S, G, $\bar{G}$, $\bar{S}$, and N represent satellite rainfall estimate, gauge rainfall measurements, mean of the gauge rainfall measurements, mean of the satellite rainfall measurements, and the number of data pairs, respectively.

| Volumetric Statistics | Formula | Unit | Best Value |
|---|---|---|---|
| Correlation Coefficient | $CC = \frac{\sum(G-\bar{G})(S-\bar{S})}{\sqrt{\sum(G-\bar{G})^2(S-\bar{S})^2}}$ | None | 1 |
| Root Mean Square Error | $RMSE = \sqrt{\left(\frac{1}{N}\right)\sum(S-G)^2}$ | mm | 0 |
| Multiplicative Bias | $Bias = \frac{\sum S}{\sum G}$ | None | 1 |

| Categorical Statistics | Formula | Unit | Best Value |
|---|---|---|---|
| Probability of Detection | $POD = \frac{A}{A+C}$ | None | 1 |
| False Alarm Ratio | $FAR = \frac{B}{A+B}$ | None | 0 |
| Critical Success Index | $CSI = \frac{A}{A+B+C}$ | None | 1 |
| Frequency Bias Index | $FBI = \frac{A+B}{A+C}$ | None | 1 |

## 3. Results and Discussion

*3.1. Model Evaluations Based on Lead Time*

3.1.1. Daily Intercomparisons

Figures 2–4 illustrate the volumetric metrics for three different lead times, LT0, LT1, and LT2, respectively, for CFSv2 and ECMWFv5.1 in comparison to daily gauge data. The volumetric metrics for LT0 are shown in Figure 2a–f. Considering the correlation coefficients at LT0, Figure 3a,d reveal that several stations in northern Nigeria, Benin, and Ghana had near-perfect temporal agreement with gauge observations (r > 0.9). The temporal agreement between gauge and the ECMWFv5.1 data in the rest of West Africa ranged between 0.2 to 0.4 and was largely > 0.2 in East and Southern Africa regions. However, the CFSv2 data showed the lowest correlation (r = −0.1) in Ethiopia compared to r = 0.4 in ECMWFv5.1 data. At LT0, stations in Kenya, Rwanda, and Burundi showed better agreement with CFSv2 than ECMWFv5.1, while the opposite was observed in Zambia and Malawi.

In terms of bias, the ECMWFv5.1 (bias < 2 mm) data consistently outperformed the CFSv2 across Africa (Figure 3b,e). The CFSv2 showed greater bias than ECMWFv5.1 along the West African coastal belt, though it largely matched the latter in the hinterland of West Africa. Again, the CFSv2 data had the highest bias (30 mm) in the Ethiopian highlands compared to 6 mm for ECMWFv5.1 data. Similarly, the CFSv2 showed consistently higher RMSE than ECMWFv5.1 across all regions but was more pronounced in Southern Africa. Overall, at LT0 the two models showed roughly the same temporal correlation pattern, but ECMWFv5.1 clearly outperformed CFSv2 due to lower bias and RMSE. This superior performance is likely linked to ECMWFv5.1's increased horizontal resolution and improvements in model configuration and initialisation, which have been shown to improve seasonal precipitation forecasts in many regions [38,48,49]. However, model performance still degrades in complex, high-elevation terrain where orographic rainfall processes are challenging to resolve; several verification studies report both improvements with ECMWFv5.1 and remaining limitations in mountainous regions [48]. The weaker performance of CFSv2 in the Ethiopian Highlands is consistent with earlier evaluations that reported moderate forecast skill for seasonal rainfall, with CFSv2 and ECMWFv5.1 correlations with observed JJA rainfall of approximately $r \approx 0.71$ and $r \approx 0.54$, respectively [50]. Such limitations are likely linked to challenges in representing complex topography and orographic rainfall processes [51]. In contrast, CFSv2 showed relatively better agreement with observations in parts of Kenya, Rwanda, and Burundi, whereas ECMWFv5.1 outperformed it in Zambia and Malawi, as documented in regional seasonal forecast evaluations [12]. Similar spatial variations in model skill across African sub-regions have been reported by Harrison et al. [21], who noted that forecast accuracy depends strongly on local rainfall regimes, topographic influences, and the representation of synoptic-scale drivers.

In West Africa, the ECMWFv5.1 correlation values are observed to be relatively better than the CFSv2. This suggests that the ECMWFv5.1 model captures rainfall patterns in this sub-region more accurately. Strong positive correlations are detected in West Africa compared to East and Southern Africa for both models. Furthermore, the bias values are generally modest for ECMWFv5.1 in most sub-regions, showing that the ECMWFv5.1 model adequately captures rainfall patterns but somewhat underestimates the gauge. In contrast, biases were often more significant for CFSv2 than ECMWFv5.1, showing that the CFSv2 model tended to overestimate rainfall in those places, notably in East Africa. It is worth noting that the RMSE values between the ECMWFv5.1 and gauge are generally low in most places, while occasional differences are visible in select stations. In contrast, the RMSE values between CFSv2 and gauge datasets are significantly greater than those of

ECMWFv5.1, implying that the CFSv2 model may not effectively reflect rainfall patterns in some regions, mainly South and East Africa.

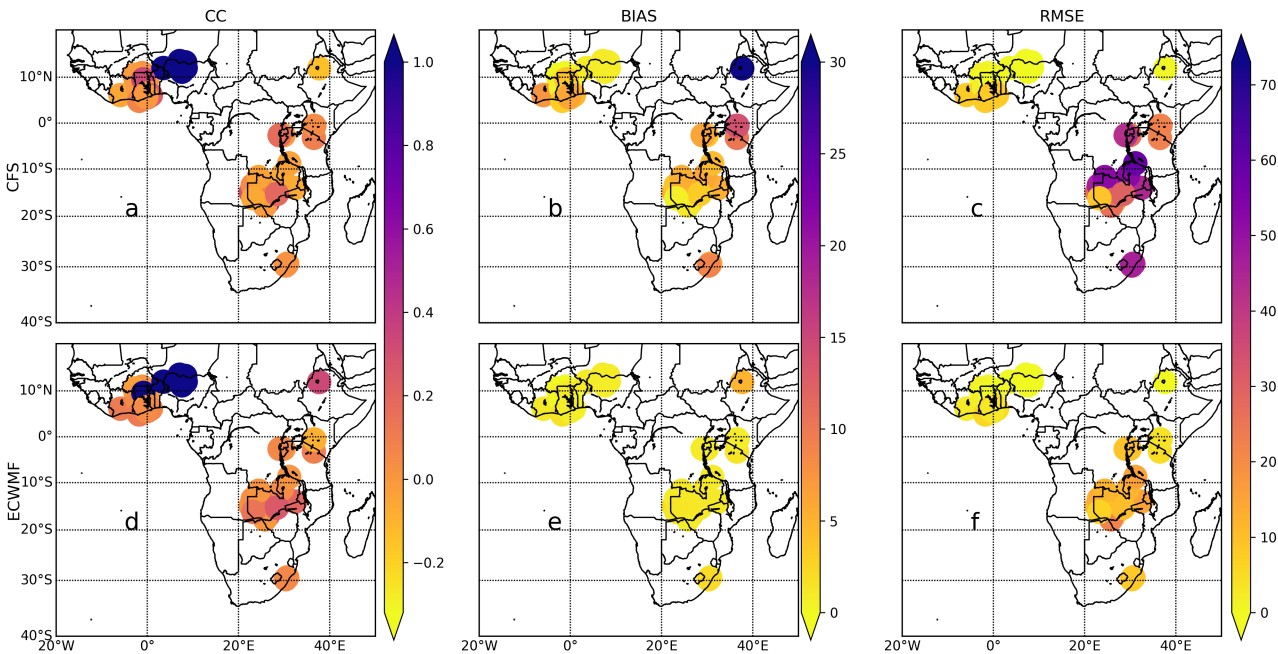

**Figure 2.** Comparison of station-level volumetric metrics between gauge observations and the model estimates at the 1-month lead time (LT0): (**a**) CC for CFSv2; (**b**) BIAS for CFSv2; (**c**) RMSE for CFSv2; (**d**) CC for ECMWFv5.1; (**e**) BIAS for ECMWFv5.1; and (**f**) RMSE for ECMWFv5.1.

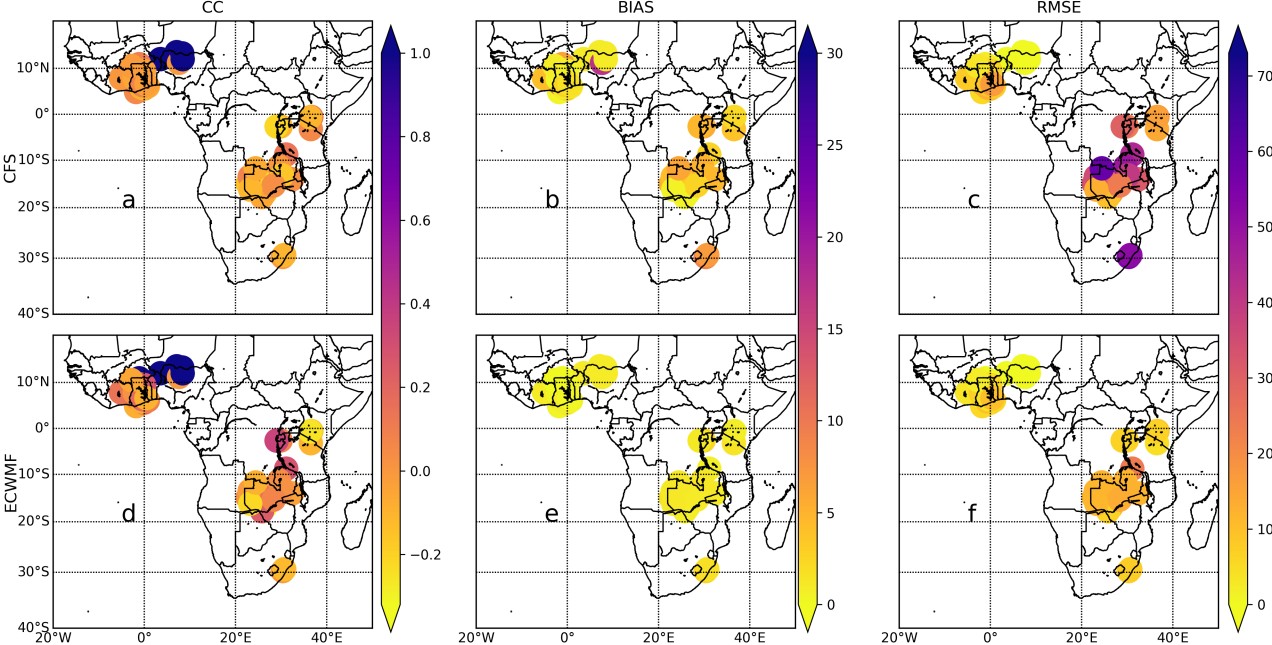

**Figure 3.** Comparison of station-level volumetric metrics between gauge observations and model estimates at the 2-month lead time (LT1): (**a**) CC for CFSv2; (**b**) BIAS for CFSv2; (**c**) RMSE for CFSv2; (**d**) CC for ECMWFv5.1; (**e**) BIAS for ECMWFv5.1; and (**f**) RMSE for ECMWFv5.1.

The volumetric measures at LT1, as shown in Figure 3, reveal that, while the correlation coefficients between the gauge and the CFSv2 model are strong, they are marginally lower than their performance at LT0 in all sub-regions. This indicates that the CFSv2 model's accuracy in capturing rainfall patterns in the region has decreased, consistent with findings

in Samala et al. [26]. Some parts of the continent (East and Southern Africa) had lower correlations, indicating that the CFSv2 model needed more accuracy in these areas as the lead time progressed. This is supported by other research that has identified similar skill drops in the CFSv2 model in certain regions [27]. Even for LT1, the ECMWFv5.1 model's correlation coefficients with the observed ones were continuously strong. For instance, in South Africa, the ECMWFv5.1 model's correlation values were typically higher than the CFSv2's. This demonstrates that the ECMWFv5.1 model may be more precise for LT1 in this region. Although high values were recorded, they were slightly lower than those obtained from LT0. The bias values show that the CFSv2 model overestimates rainfall in most portions of Africa, particularly in East Africa, similar to what was observed in LT0. The ECMWFv5.1 model's values follow a similar trend to its performance in LT0, with small underestimations noted. The RMSE values for CFSv2 remained relatively high in South Africa, with a minor increase in some areas of West Africa. RMSE values for ECMWFv5.1 were at the same margins in all sub-regions as in LT0. The consistency of ECMWFv5.1 performance across multiple lead times suggests robustness to lead-time degradation that has been specifically documented in earlier valuations [38], corroborated by ECMWFv5.1's operational forecast performance summaries [52], and observed across the sub-seasonal to seasonal project's multi-model predictions [53].

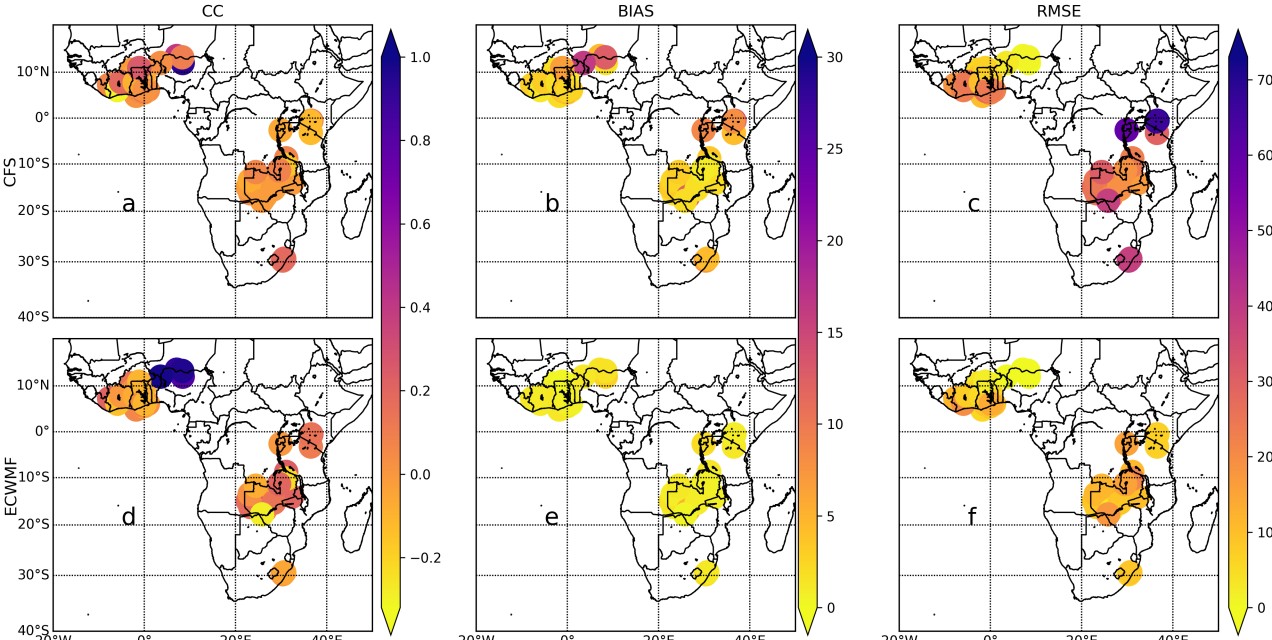

**Figure 4.** Comparison of station-level volumetric metrics between gauge observations and model estimates at the 3-month lead time (LT2): (**a**) CC for CFSv2; (**b**) BIAS for CFSv2; (**c**) RMSE for CFSv2; (**d**) CC for ECMWFv5.1; (**e**) BIAS for ECMWFv5.1; and (**f**) RMSE for ECMWFv5.1.

Figure 4 illustrates the volumetric metrics at LT2. The range of correlation coefficients obtained has reduced considerably further compared to what was seen in LT1, with most stations having lower r values. As the number of lead periods increased, the CFSv2's correlation with the observed rainfall data at most sites began to decline. For ECMWFv5.1, we observe that correlation coefficients are generally higher than those of the CFSv2 in most sub-regions. Although both datasets show a decline in correlation values at different stations as the lead times increased, ECMWFv5.1 still produced higher r values than CFSv2. The bias values between observed and CFSv2 data increased significantly in all sub-regions, particularly for stations in West Africa. Observing ECMWFv5.1 biases at LT2 (Figure 4e), we see virtually the same margin of error as in LT1 across Africa. The RMSE values for CFSv2

have continued to rise in all sub-regions, particularly in East Africa. The ECMWFv5.1 RMSE scale values are small over Africa, but we detected a few occasions when the values increased slightly. On average, ECMWFv5.1 significantly outperformed CFSv2 in all sub-regions, especially in West Africa (Table 4). Although correlation coefficients for CFSv2 in East and Southern Africa were relatively high, their error margins were generally much higher than those of the ECMWFv5.1. This continued superiority of ECMWFv5.1 across lead times aligns with past studies linking its skill to more advanced ensemble generation and better handling of tropical convection [54], as well as similar early warning forecast verification work by Harrison et al. [21]. The progressive decline in CFSv2 accuracy may reflect broader issues with its representation of tropical convection and associated mean-state biases—documented to impair forecast skill and convection propagation in sub-seasonal prediction [55].

**Table 4.** Average categorical and volumetric statistical metrics for the ECMWFv5.1 and CFSv2 with respect to gauge observations at the three sub-regions on a daily time scale. The metrics are probability of detection (POD), critical success index (CSI), false alarm ratio (FAR), and frequency of bias index (FBI), respectively. R, BIAS, and RMSE are the volumetric metrics that represent correlation coefficient, multiplicative bias, and root mean square error, respectively.

| | CFSv2 | | | | | | | ECMWFv5.1 | | | | | | |
|---|---|---|---|---|---|---|---|---|---|---|---|---|---|---|
| | POD | FAR | BIAS | CSI | FBI | R | RMSE | POD | FAR | BIAS | CSI | FBI | R | RMSE |
| West Africa | 0.86 | 0.20 | 1.80 | 0.73 | 1.67 | 0.62 | 1.38 | 0.93 | 0.13 | 0.90 | 0.80 | 2.20 | 0.80 | 0.87 |
| South Africa | 0.88 | 0.38 | 3.49 | 0.57 | 1.50 | 0.31 | 27.00 | 0.96 | 0.42 | 1.43 | 0.57 | 1.74 | 0.20 | 12.06 |
| East Africa | 0.93 | 0.57 | 8.12 | 0.43 | 3.07 | 0.10 | 32.00 | 0.96 | 0.59 | 1.22 | 0.40 | 3.80 | 0.10 | 7.05 |
| Africa | 0.89 | 1.15 | 4.47 | 0.57 | 2.08 | 0.34 | 20.01 | 0.95 | 0.38 | 1.18 | 0.59 | 2.58 | 0.37 | 6.66 |

In Figures 5–7, we show the results for the categorical metrics of POD, CSI, FAR, and FBI for the three different lead times; LT0, LT1, and LT2, respectively, for CFSv2 and ECMWFv5.1 in comparison to gauge data. For LT0 we observe that across South Africa, ECMWFv5.1 showed superior rainfall detection skill (POD of about 1) for all stations compared to CFSv2 (see Figure 5a,e). The high POD values observed for ECMWFv5.1 were again demonstrated across West and East Africa, but to a relatively lesser degree. This indicates that ECMWFv5.1 generally could detect rainfall events compared to observation in all sub-regions correctly. Similarly, CFSv2 has good rainfall detection skills but is relatively lower than ECMWFv5.1 in all sub-regions. Specifically, CFSv2 has a lower rainfall detection skill in West Africa than in the other sub-regions. The POD values for CFSv2 are 0–1 to 0.8–1 to 0.25–1 in West, East, and South Africa, respectively. This indicates that CFSv2 had the highest POD in east Africa. Both CFSv2 and ECMWFv5.1 are seen to have remarkably similar detection strengths in East Africa. On average, ECMWFv5.1 detects more rainfall in South and West Africa than CFSv2. In South Africa, CSI values were frequently higher for ECMWFv5.1 than CFSv2, similar to the POD values. Yet, for East Africa, the CSI values for both products were virtually invariably the same. While CFSv2 and ECMWFv5.1 had false rainfall detection (FAR) in West Africa's westernmost regions, they also had the lowest FAR in the sub-region's easternmost portions. FARs for both products in each sub-region were generally similar. The FBI values for CFSv2 and ECMWFv5.1 in West Africa ranged from 0–7 to 0–12, respectively. In South and East Africa, FBI values for both ECMWFv5.1 and CFSv2 were comparable, with most >1, indicating overforecasting. These results for LT0 align with previous findings that forecast skill tends to decrease in complex topographic and coastal regions, highlighting the advantage of ECMWFv5.1 in reducing false alarms and improving rainfall detection accuracy.

At LT1 (Figure 6), we see that the POD values for West Africa observed by both CFSv2 and ECMWFv5.1 range between 0 and 1. Nonetheless, when compared to LT0, a majority of the stations in this sub-region experienced a slight drop in POD values, which is consistent with the expected decline in forecast skill with longer lead times. Similarly, we observe a drop in POD for CFSv2 in East Africa. However, in East and South Africa, the POD values for ECMWFv5.1 for LT1 are fairly comparable to those of LT0. For West Africa, the CSI values for CFSv2 and ECMWFv5.1 range from 0 to 1. Yet, compared to LT0, most of the stations in this sub-region saw a drop in CSI values. Similarly, CSI for CFSv2 and ECMWFv5.1 has reduced in East Africa. Nonetheless, the CSI values for CFSv2 and ECMWFv5.1 in South Africa were reasonably close for LT1 and LT0. CFSv2 and ECMWFv5.1 detected a slight increase in the FAR across all sub-regions. Within West Africa, ECMWFv5.1 had a somewhat larger frequency of bias index than CFSv2. However, this was generally minimal and did not significantly affect the detection accuracy of ECMWFv5.1 compared to CFSv2.

At LT2 (Figure 7), the POD values observed by CFSv2 were further reduced across all African sub-regions. Conversely, the POD values for ECMWFv5.1 slightly rose for most sites in West Africa while remaining relatively steady in East and South Africa. In West Africa, the CSI values observed by CFSv2 and ECMWFv5.1 vary from 0 to 0.6, indicating a reduction in CSI as the lead time progressed. Within South Africa, a similar trend in CSI is found for both products. Nonetheless, the CSI for LT2 in East Africa is shown to be relatively good for ECMWFv5.1. The FARs for both products were noted to rise in West and South Africa compared to LT1 and LT0. Furthermore, the FBI in West Africa declined significantly, as detected by ECMWFv5.1, while being almost the same in East and South Africa for both CFSv2 and ECMWFv5.1. Overall, these LT2 results reinforce the earlier observation that ECMWFv5.1 consistently outperforms CFSv2 in most sub-regions, particularly in West Africa, even as lead times increase—a finding supported by similar comparative studies in the seasonal forecasting literature.

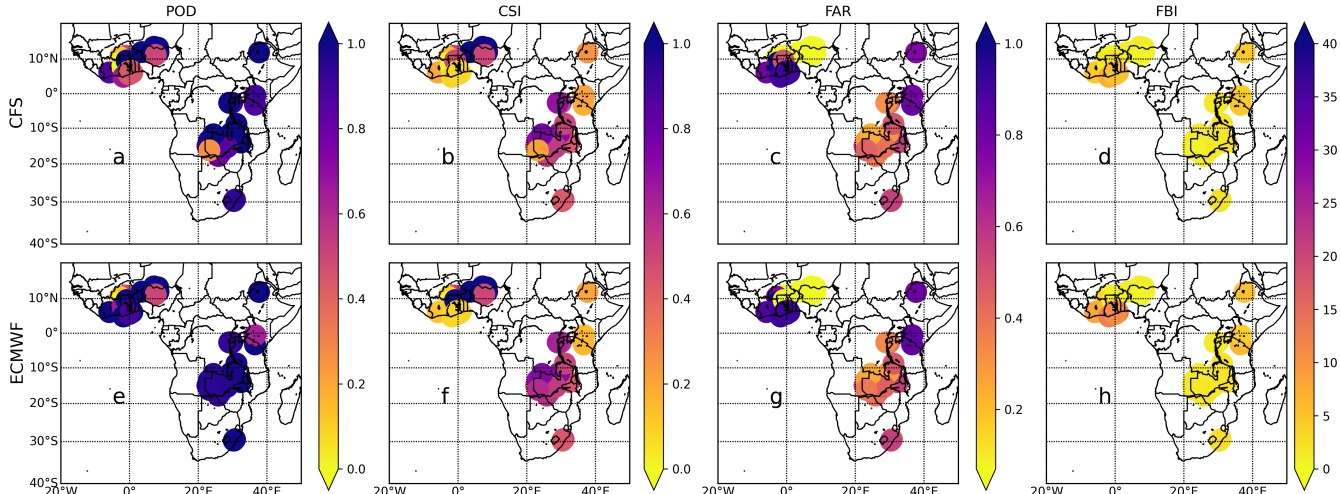

**Figure 5.** Comparison of station-level categorical metrics between gauge observations and model estimates at the 1-month lead time (LT0): (**a**) POD for CFSv2; (**b**) CSI for CFSv2; (**c**) FAR for CFSv2; (**d**) FBI for CFSv2; (**e**) POD for ECMWFv5.1; (**f**) CSI for ECMWFv5.1; (**g**) FAR for ECMWFv5.1; and (**h**) FBI for ECMWFv5.1.

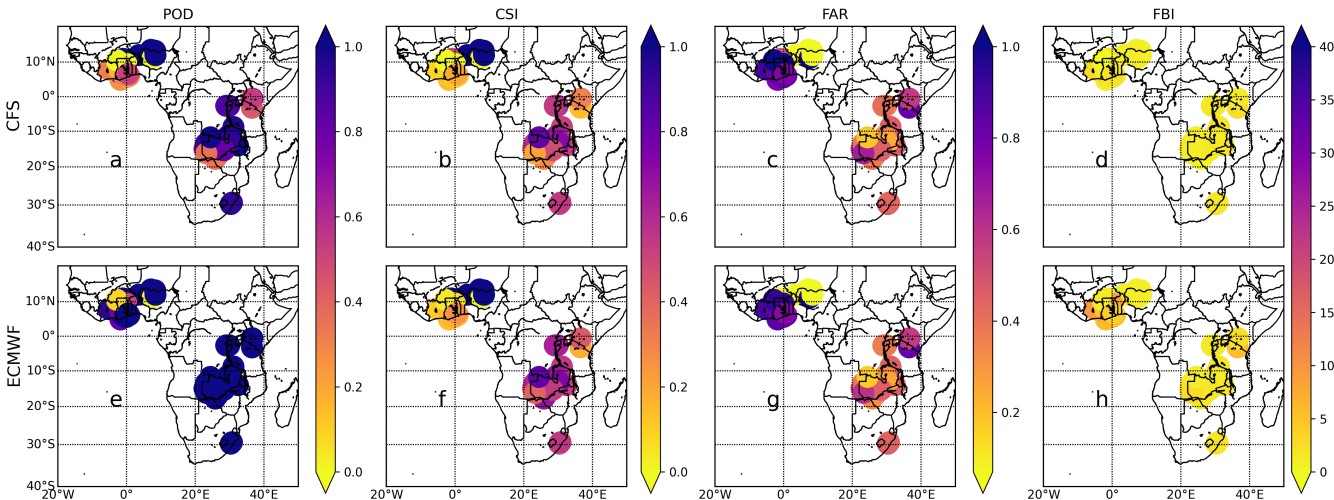

**Figure 6.** Comparison of station-level categorical metrics between gauge observations and model estimates at the 2-month lead time (LT1): (**a**) POD for CFSv2; (**b**) CSI for CFSv2; (**c**) FAR for CFSv2; (**d**) FBI for CFSv2; (**e**) POD for ECMWFv5.1; (**f**) CSI for ECMWFv5.1; (**g**) FAR for ECMWFv5.1; and (**h**) FBI for ECMWFv5.1.

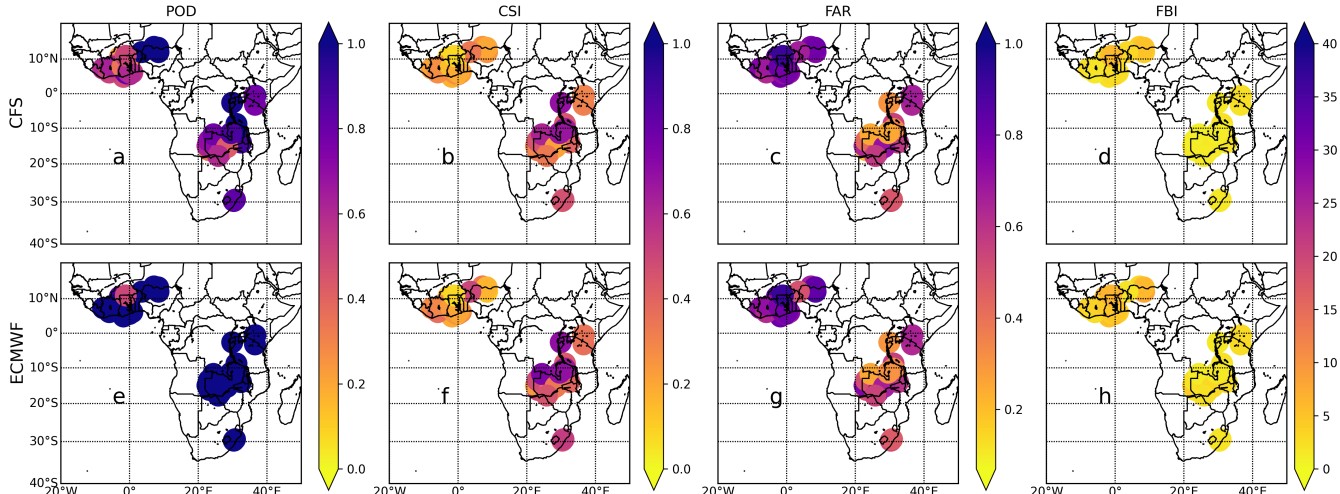

**Figure 7.** Comparison of station-level categorical metrics between gauge observations and model estimates at the 3-month lead time (LT2): (**a**) POD for CFSv2; (**b**) CSI for CFSv2; (**c**) FAR for CFSv2; (**d**) FBI for CFSv2; (**e**) POD for ECMWFv5.1; (**f**) CSI for ECMWFv5.1; (**g**) FAR for ECMWFv5.1; and (**h**) FBI for ECMWFv5.1.

According to the results, the CFSv2 and ECMWFv5.1 models performed better at LT0 than LT1 and LT2, on average. However, the outcomes for LT0 and LT1 were somewhat comparable. A significance test was performed on the measures to assess whether LT0 significantly outperformed LT1. Considering that a 99 percent confidence interval was used, the significance threshold was 1 percent. Figures 8 and 9 show the *p*-values for the difference between LT0 and LT1 for volumetric and categorical metrics. The figures demonstrate that the *p*-values for all metrics are extremely small (less than 0.01), indicating that LT0 significantly outperformed LT1. In Figure 8, the bias of CFSv2 as seen in the near-flat trend line occurs because the analyzed region exhibits little long-term change in the variable considered over the study period, likely due to relatively stable climatic conditions and compensating variations within sub-regions. This was a clear example of how LT0 and LT1 differed, and thus the superior performance of LT0. Consequently, LT0 was used for all subsequent analyses.

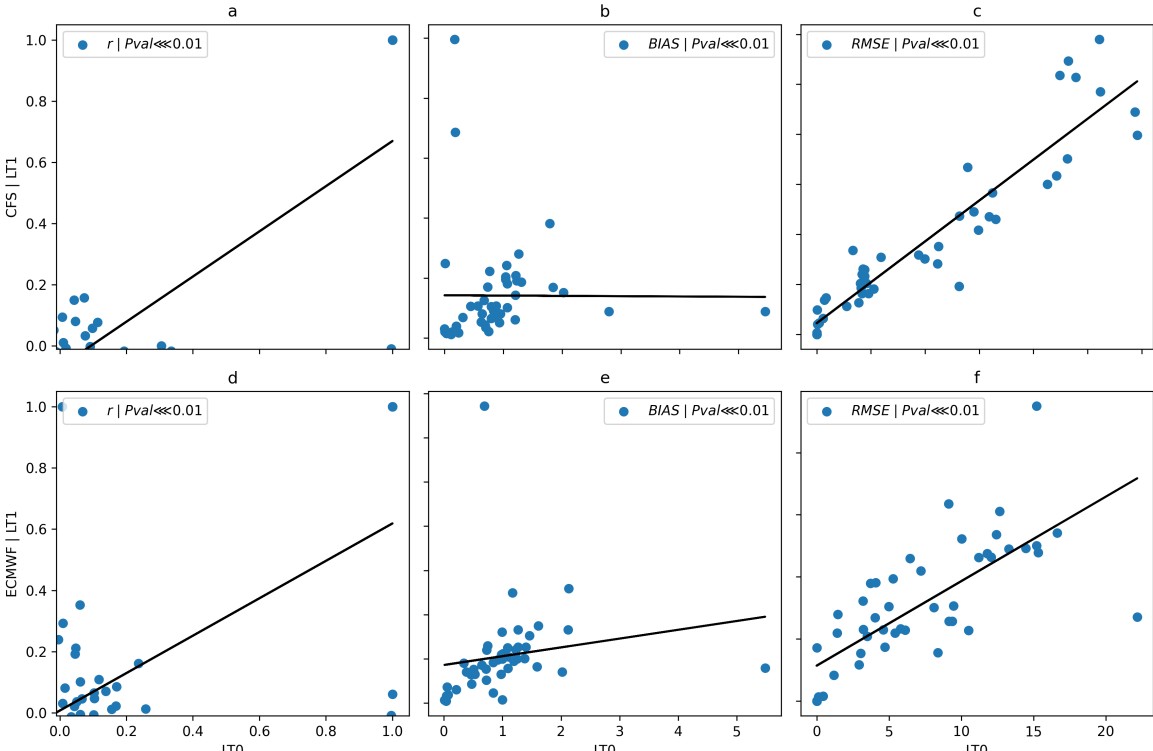

**Figure 8.** Significance test showing the p-values (Pval) at a confidence level of 99% and a significance value of 0.01 between LT0 and LT1 for both models. (**a**) Pval for CC for CFSv2; (**b**) Pval for BIAS for CFSv2; (**c**) Pval for RMSE for CFSv2; (**d**) Pval for CC for ECMWFv5.1; (**e**) Pval for BIAS for ECMWFv5.1; and (**f**) Pval for RMSE for ECMWFv5.1. Bold black line is the line of best fit, indicating whether there is a significant difference between the two lead times (Pval < 0.01) or not (Pval > 0.01).

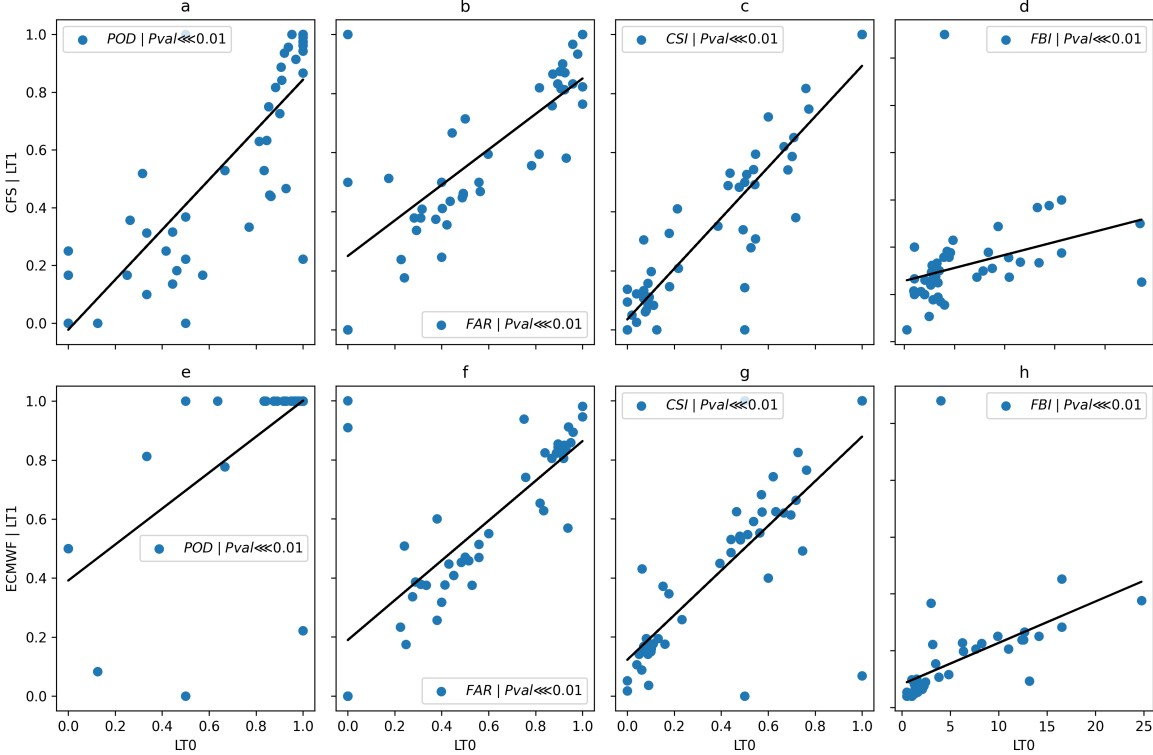

**Figure 9.** Significance test showing the p-values (Pval) at a confidence level of 99% and a significance value of 0.01 between LT0 and LT1 for both models based on categorical metrics. (**a**) Pval for POD for

CFSv2; (**b**) Pval for CSI for CFSv2; (**c**) Pval for FAR for CFSv2; (**d**) Pval for FBI for CFSv2; (**e**) Pval for POD for ECMWFv5.1; (**f**) Pval for CSI for ECMWFv5.1; (**g**) Pval for FAR for ECMWFv5.1; and (**h**) Pval for FBI for ECMWFv5.1. Bold black line indicates whether there is a significant difference between the two lead times (Pval < 0.01) or not (Pval > 0.01).

3.1.2. Dakadal Intercomparisons of Model Performances

The performance of the models with respect to gauge at dekadal (10 days mean) time scale is evaluated in this section. Figure 10 displays the findings of the dekadal analysis for volumetric metrics across Africa. The results shows that the CC values for CFSv2 are fairly good across the region (>0.4), with a few exceptions in South, West, and East Africa (Figure 10a,d). The ECMWFv5.1 had the highest CC values of the two, with almost all stations having a CC value greater than 0.5. The CC values for ECMWFv5.1 were higher than those for CFSv2 in West Africa, as a strong positive association was found between the gauge data and the ECMWFv5.1. Although relatively high values for CFSv2 were seen throughout Africa, the bias values for both models were relatively small. Most notably, in East and South Africa, we saw relatively high RMSE values for CFSv2, while West Africa had the lowest values, indicating good skill of CFSv2 in the sub-region. Besides a few small spikes in East and South Africa with values no higher than 40 mm, ECMWFv5.1 showed modest RMSE values across Africa. ECMWFv5.1 values for West Africa were also better than those for other sub-regions.

Figure 11 depicts the dekadal categorical metrics for CFSv2 and ECMWFv5.1 over Africa's sub-regions. CFSv2 and ECWMF-S5 were found to have greater PODs in East and South Africa (POD of approximately 1). POD values measured by CFSv2 and ECMWFv5.1 in West Africa are 0–1 and 0.5–1, respectively. This demonstrates ECMWFv5.1's ability to detect more dekadal rainfall occurrences than CFSv2 within West Africa. The CSI values of CFSv2 and ECMWFv5.1 were relatively comparable within East Africa, ranging from 0.4 to 1 for both models. A comparable CSI was also obtained for both datasets within South Africa. Yet, throughout West Africa, the CSI values for CFSv2 and ECMWFv5.1 ranged from 0 to 1 and 0.25 to 1, respectively. As a result, both datasets showed that South Africa exhibited relatively high CSI, followed by East Africa and West Africa.

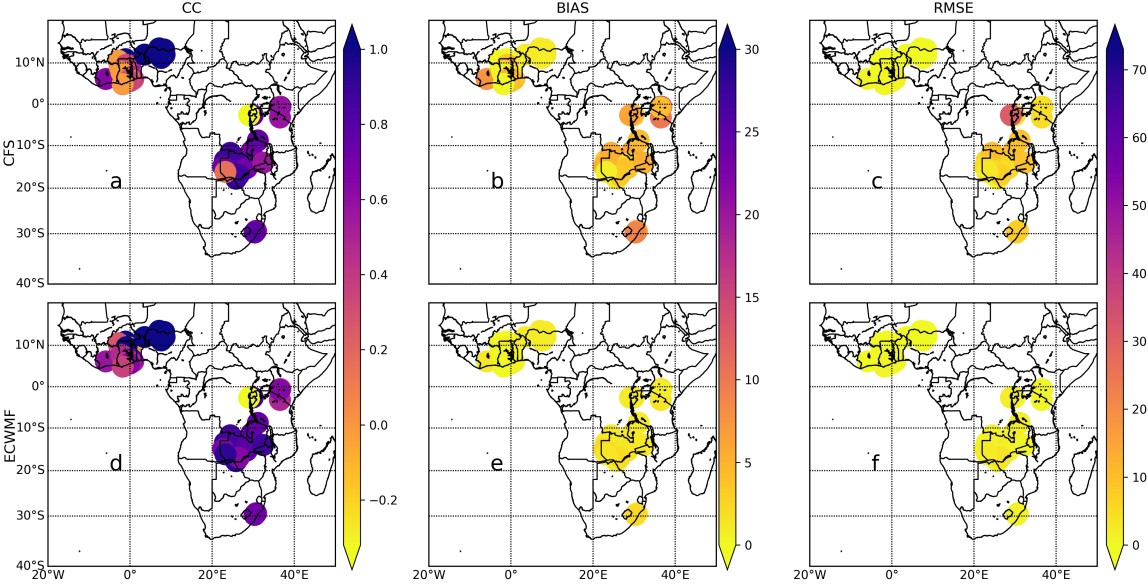

**Figure 10.** Comparison of station-level volumetric metrics between gauge observations and model estimates at the dekadal time scale. (**a**) CC for CFSv2; (**b**) BIAS for CFSv2; (**c**) RMSE for CFSv2; (**d**) CC for ECMWFv5.1; (**e**) BIAS for ECMWFv5.1; and (**f**) RMSE for ECMWFv5.1.

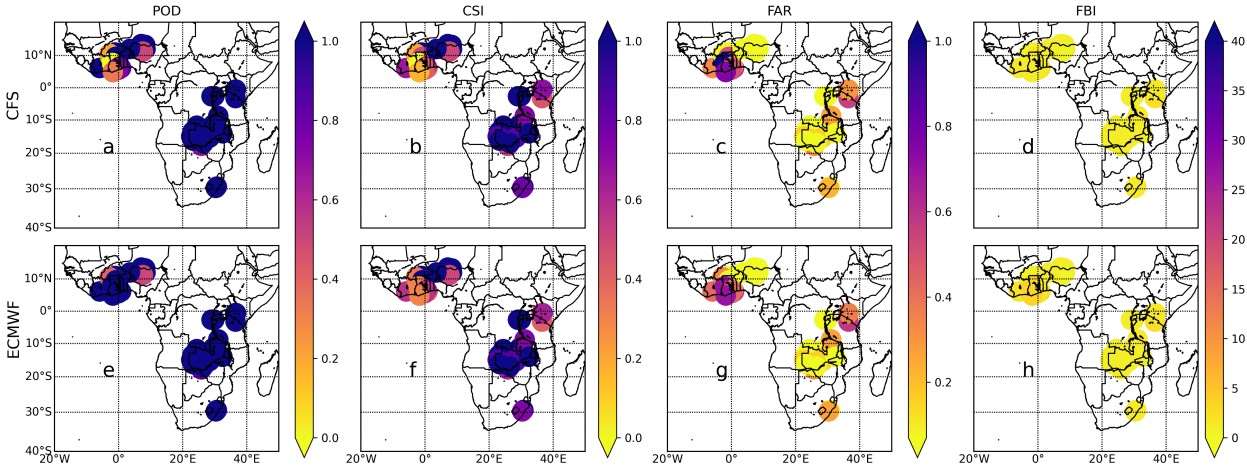

**Figure 11.** Comparison of station-level categorical metrics between gauge observations and model estimates at the dekadal time scale. (**a**) POD for CFSv2; (**b**) CSI for CFSv2; (**c**) FAR for CFSv2; (**d**) FBI for CFSv2; (**e**) POD for ECMWFv5.1; (**f**) CSI for ECMWFv5.1; (**g**) FAR for ECMWFv5.1; and (**h**) FBI for ECMWFv5.1.

Within West Africa, the FAR values range from 0 to 1 and from 0 to 0.7 for CFSv2 and ECMWFv5.1, respectively. This demonstrates that CFSv2 observed higher FAR within West Africa than ECMWFv5.1. Nonetheless, the FAR recorded by both datasets within East and South Africa were quite comparable. Consequently, within East and South Africa, FAR ranged from 0 to 0.6 and 0 to 0.3, respectively. As a result, both datasets showed very low FAR within South Africa, followed by East and West Africa. This illustrates that CFSv2 and ECMWFv5.1 detected more erroneous dekadal rainfall in West Africa than in East and South Africa. The FBI values for the entire continent were modest (<2 mm) for both models, showing that the models performed better at dekadal time frames (Table 5).

Over dekadal time scales, ECMWFv5.1 outperforms CFSv2 in all sub-regions, depicting the model's capacity to reflect the rainfall pattern over the continent better. Both models may be best suited for hydrological studies in West Africa. Generally, while product performance declines with rising lead times, it improves with greater temporal resolutions.

**Table 5.** Average categorical and volumetric statistical metrics for ECMWFv5.1 and CFSv2 with respect to gauge observations at the three sub-regions on a dekadal time scale. The metrics are probability of detection (POD), critical success index (CSI), false alarm ratio (FAR), and frequency of bias index (FBI), respectively. R, BIAS, and RMSE are the volumetric metrics that represent correlation coefficient, multiplicative bias, and root mean square error, respectively.

| | **CFSv2** | | | | | | | **ECMWFv5.1** | | | | | | |
| --- | --- | --- | --- | --- | --- | --- | --- | --- | --- | --- | --- | --- | --- | --- |
| | POD | FAR | BIAS | CSI | FBI | R | RMSE | POD | FAR | BIAS | CSI | FBI | R | RMSE |
| West Africa | 0.98 | 0.14 | 1.82 | 0.86 | 1.23 | 0.80 | 0.24 | 0.93 | 0.06 | 0.90 | 0.87 | 1.04 | 0.92 | 0.08 |
| East Africa | 1.00 | 0.28 | 8.12 | 0.72 | 1.63 | 0.57 | 17.92 | 1.00 | 0.29 | 1.22 | 0.71 | 1.69 | 0.50 | 1.36 |
| South Africa | 1.00 | 0.04 | 3.50 | 0.96 | 1.05 | 0.82 | 8.32 | 1.00 | 0.08 | 1.43 | 0.92 | 1.09 | 0.75 | 2.39 |
| Africa | 0.99 | 0.15 | 4.48 | 0.85 | 1.30 | 0.73 | 8.83 | 0.98 | 0.14 | 1.19 | 0.83 | 1.27 | 0.72 | 1.28 |

### 3.2. Performances of Models in Changing Elevations

Figure 12 describes the performance of the models at various elevations. Figure 13 depicts the mean modified Kling–Gupta Efficiency (KGE′) scores for the CFSv2 and ECMWFv5.1 models at various points. KGE′ values for CFSv2 and ECWMF-S5, respectively, were determined to be in the range of 0–0.9 and 0.6–1 for L1. At this height, the ECWMF-S5 model exhibited relatively high KGE′ values, indicating that it performed better than the

CFSv2 model. Despite this, both models performed well, except for a few CFSv2 stations with a zero KGE′ score. Nonetheless, the ECMWFv5.1 model has substantially greater skill than CFSv2. This is supported by correlation values of >0.90 for ECMWFv5.1 and <0.90 for CFSv2, respectively. Although both models' KGE′ values at L2 had negative-score lower bounds, it was found that ECMWFv5.1 performed relatively better than CFSv2 (Figure 12a). The KGE′ values for L3–L4 were negative, suggesting poor model performance at these heights. CFSv2 is shown to be relatively more skilled at L3 and L4. The $\beta$ values for ECMWFv5.1 at L1 were found to be approximately 1, whereas CFSv2 had a value > 1. This means that ECMWFv5.1 has nearly perfect skill, but CFSv2 is observed to overestimate the gauge slightly. For L2–L4, we observe very high $\beta$ values for CFSv2, indicating significant overestimation from the model. The $\gamma$ values were close to one for both models, although slightly higher for ECMWFv5.1, indicating strong performance at L1. These values are significantly reduced with higher elevations (L2–L4). Generally, both models are excellent at capturing rainfall at low elevations (<500 m), but ECMWFv5.1 outperforms CFSv2. Both models perform poorly at high heights >1000 m, although the CFSv2 shows a slighly better skill, highlighting the models' limitations at higher elevations. In general, both models at L1 have minimal error margins and excellent detection and critical success indices. Both models' correlation coefficients were high; however, ECMWFv5.1 had significantly higher values than CFSv2. At L2–L4, we notice a decline in POD, CSI, and r values while observing large but significantly greater error margins for CFSv2 than ECMWFv5.1. This supports earlier findings, which show that the models perform better at lower elevations and less efficiently at higher elevations (Table 6).

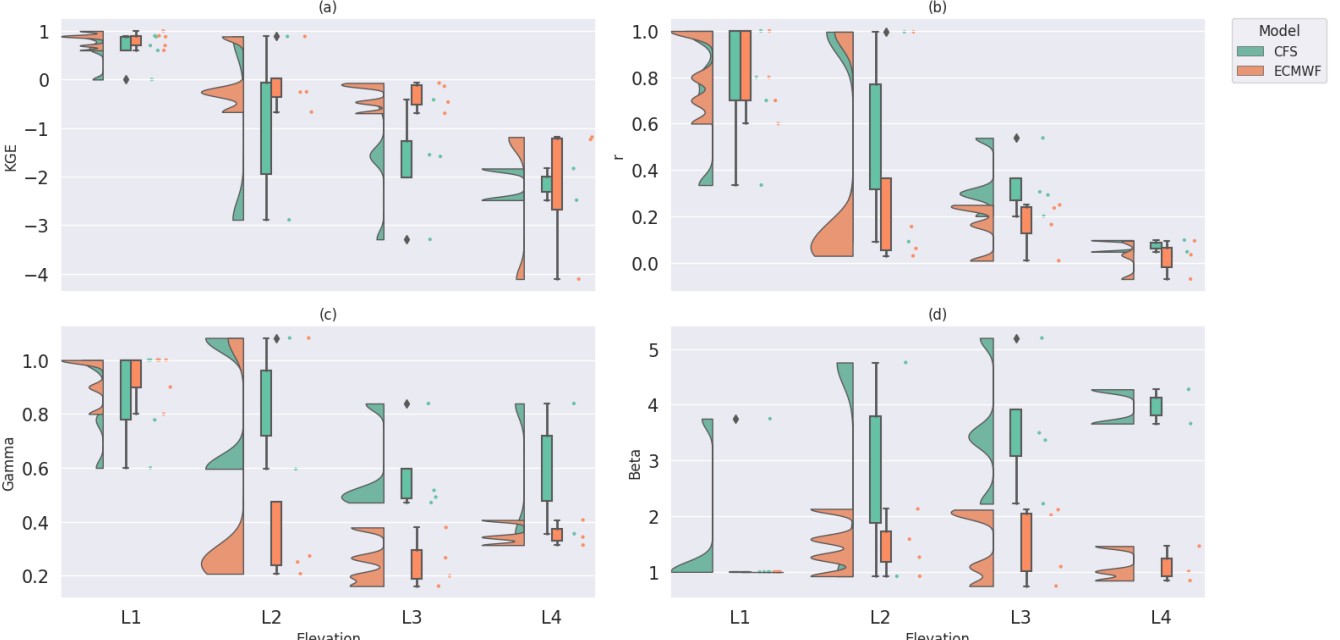

**Figure 12.** Raincloud plots showing the performance of the products at various elevations from 2012 to 2022. (**a**) KGE′; (**b**) correlation coefficient (r); (**c**) variability ($\gamma$); and (**d**) frequency bias ($\beta$). Elevation ranges are L1 (<500 mm), L2 (500–1000 mm), L3 (1000–1500 mm), and L4 (>1500 mm).

**Table 6.** Average categorical and volumetric statistical metrics for ECMWFv5.1 and CFSv2 with respect to gauge observations at various elevations. Elevation ranges are L1 (<500 mm), L2 (500–1000 mm), L3 (1000–1500 mm), and L4 (>1500 mm).

| | CFSv2 | | | | | | | ECMWFv5.1 | | | | | | |
|---|---|---|---|---|---|---|---|---|---|---|---|---|---|---|
| | POD | FAR | BIAS | CSI | FBI | R | RMSE | POD | FAR | BIAS | CSI | FBI | R | RMSE |
| L1 | 1 | 0.1 | 1.55 | 0.9 | 1.2 | 0.87 | 0.04 | 1 | 0 | 1 | 1 | 1 | 0.98 | 0.01 |
| L2 | 0.82 | 0.37 | 5.23 | 0.49 | 1.51 | 0.28 | 30.07 | 0.86 | 0.38 | 1.48 | 0.49 | 1.64 | 0.31 | 6.67 |
| L3 | 0.89 | 0.37 | 3.57 | 0.58 | 1.52 | 0.33 | 26.30 | 0.96 | 0.40 | 1.49 | 0.57 | 1.74 | 0.16 | 12.57 |
| L4 | 0.88 | 0.64 | 6.02 | 0.35 | 3.47 | 0.05 | 29.48 | 0.95 | 0.65 | 1.11 | 0.34 | 4.13 | 0.02 | 9.50 |

### 3.3. Evaluation of Models at the Three Sub-Regions

The performance of the models at various sub-regions was evaluated using the modified Kling–Gupta efficiency (KGE′) statistic measure. The raincloud plots in Figure 13 display the results for ECMWFv5.1 and CFSv2. While negative KGE values represent "bad" model simulations, positive KGE levels reflect "good" model simulations. Both models performed better in the West Africa and East Africa regions, though ECMFW was more skillful. In West Africa ECMWFv5.1 mimicked the variability of gauge rainfall compared to other regions where huge overestimations were recorded. The poor performance of the models in East Africa region could be attributed to topographical heterogeneity. The accuracy of predicting rainfall is lower in complex mountainous landscapes in West Africa [24].

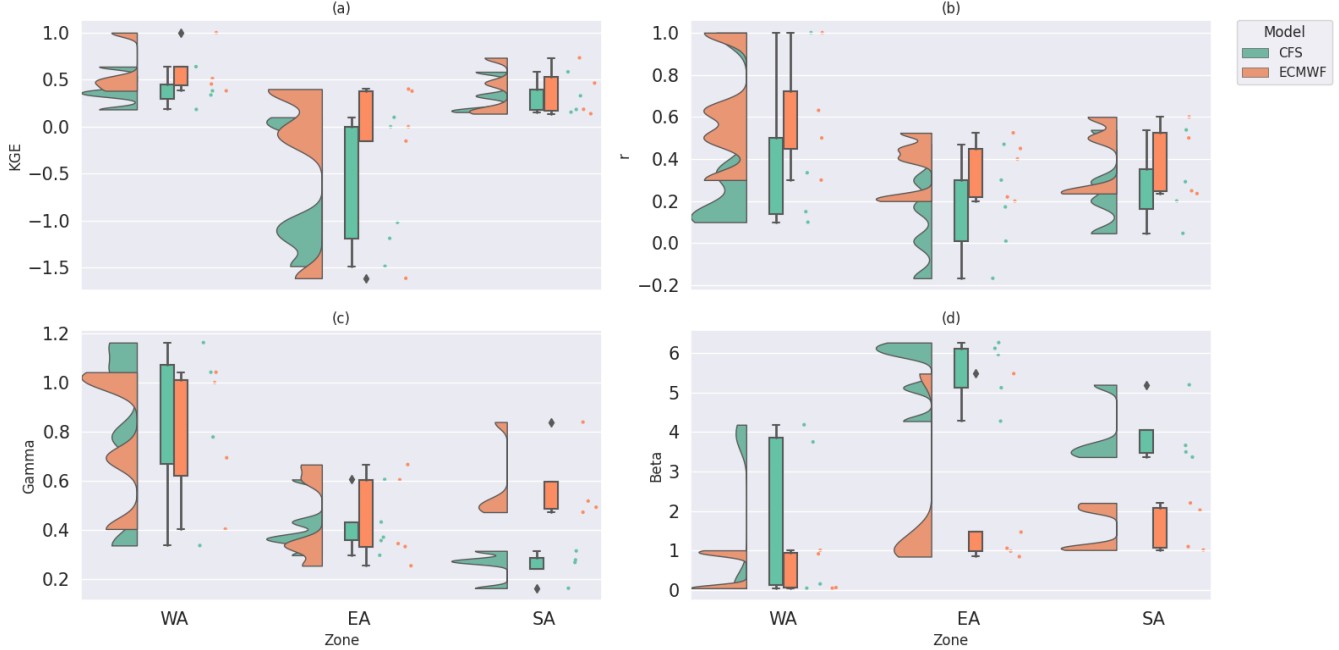

**Figure 13.** Raincloud plots showing the performance of the products at the three sub-regions from 2012 to 2022. (**a**) KGE′; (**b**) correlation coefficient (r); (**c**) variability ($\gamma$); and (**d**) frequency bias ($\beta$).

The cumulative distribution function (CDF) for CFSv2 and ECMWFv5.1 for Africa and its three sub-regions is shown in Figure 14. CFSv2 generally overestimated low-intensity rainfall events (<20 mm) and high-intensity rainfall events (>20 mm) across Africa (i.e., the CDF curve for CFSv2 is below the observation curve). While ECMWFv5.1 somewhat overestimated low-intensity rainfall, it was also shown to have a nearly comparable distribution to the gauge, especially for high-intensity rain. This demonstrates that CFSv2 overestimates low- and high-intensity rainfall events over Africa, while ECMWF slightly underestimates low-intensity rainfall events while accurately capturing high-intensity events. Both mod-

els were shown to replicate low-intensity rainfall effectively in West Africa, with a slight overestimation. Both models are revealed to capture high-intensity rain. CFSv2 overestimated both low- and high-intensity rainfall in South and East Africa. The ECMWFv5.1 model, on the other hand, is found to overestimate low-intensity rainfall while accurately predicting high-intensity rainfall. The CFSv2 model is observed to have weak $\gamma$ values ($<0.3$) compared to the ECMWFv5.1 model ($> 0.5$), indicating a better performance of ECMWFv5.1.

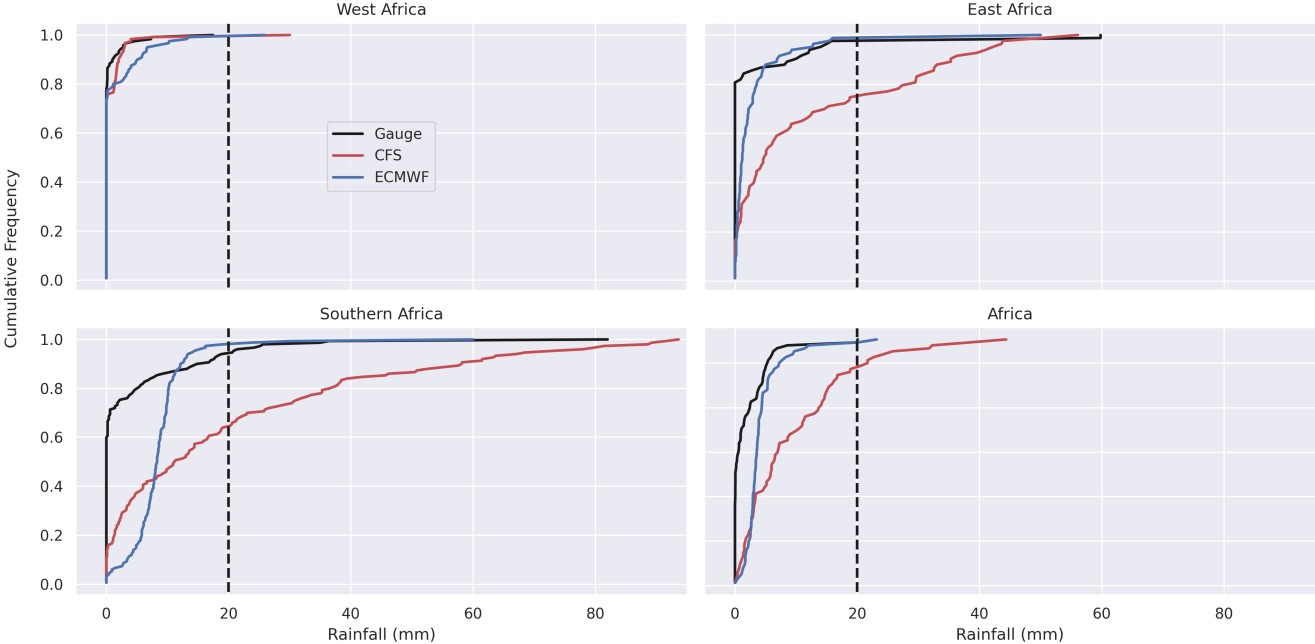

**Figure 14.** Inter-comparison of cumulative distribution functions of the gauge observations and model rainfall estimates (CFSv2 and ECMWFv5.1) for the 1-month lead time (LT0) over Africa and the three sub-regions. Subfigures correspond to: West Africa (**top-left**), East Africa (**top-right**), Southern Africa (**bottom-left**), and Africa average (**bottom-right**). The dotted vertical line at 20 mm is the cutoff used to categorize low ($<20$ mm) and high ($>20$ mm) rainfall events.

## 4. Conclusions

The present work evaluated the relative performance of the ECMWFv5.1 and the CFSv2 gridded rainfall models across three regions in Africa from 2012–2022. The validation uses three strategies: lead-time-based, regional (spatial), and elevation gradients. The results reveal that the model's accuracy to capture rainfall decreases with increasing lead times, but it improves with higher temporal resolutions. At all lead times, ECMWFv5.1 gives a better reflection of observed rainfall than the CFSv2 data due to more significant correlation coefficients, minimal bias, and RMSE values. It was also discovered that the CFSv2 model overestimates precipitation in most parts of Africa, particularly in East Africa. The ECMWFv5.1 model, on the other hand, had a few minor underestimations. In West Africa, both models perfomed well; however, the CFSv2 has a lower rainfall detection score. On dekadal timescales, ECMWFv5.1 outperformed CFSv2 in all sub-regions, illustrating the potential of ECMWFv5.1 to capture the rainfall pattern throughout the continent better. On both daily and dekadal time scales, the ECMWFv5.1 model performed better in the South and West Africa. Generally, both models are excellent at capturing rainfall at low elevations ($<500$ m), but ECMWFv5.1 outperforms CFSv2, and both models perform poorly at high elevations $>500$ mm. ECMWFv5.1 significantly outperformed CFSv2 in all sub-regions, especially in West Africa. Although the correlation coefficients for CFSv2 in South Africa were fairly good, their error margins were generally much higher than

those of the ECMWFv5.1. In general, we observed that CFSv2 overestimates observations, whereas ECMWFv5.1 underestimates observations. It was observed that CFSv2 tends to overestimate low- and high-intensity rainfall events. At the same time, the ECMWFv5.1 slightly underestimates low-intensity rainfall events and accurately captures high-intensity events over Africa. Overall, the accuracy of these models in predicting rainfall patterns in Africa varies according to the lead time, region, intensity of rainfall, and elevation.

One limitation of this study is the sparse and uneven distribution of rain gauge stations across Africa, which served as the reference data for validation. In particular, the number of stations with high-quality data was 29 in West Africa, 19 in Southern Africa, but only 4 in East Africa. This imbalance reduces the ability to fully capture regional variability, especially in the complex terrain of East Africa, and may introduce biases in continent-wide conclusions. Therefore, while the models demonstrate strong potential, results from underrepresented regions should be interpreted with caution. While more gauge data were available especially in East Africa, the quality of data was low and hence could not be utilized. This portrays the general trends across Africa where available rain gauge networks are sparse and declining, and their records are characterized by many gaps (12). This highlights the need for investing in modern automated observation networks like the initiative promoted by the Trans-African Hydro-Meteorological Observatory [56]. We recommend bias-correction in future studies to improve the agreement between rainfall observations and the forecasts.

As a result, it is critical to continue updating and refining these models to improve their accuracy and reliability for application in various industries. This study emphasized the potential of the CFSv2 and ECMWFv5.1 models for climate impact studies, particularly in West Africa and other low-elevation regions.

**Author Contributions:** Conceptualization, W.A.A., E.G.B. and F.K.M.; methodology, W.A.A., E.G.B. and F.K.M.; software, W.A.A., E.G.B. and F.K.M.; validation, W.A.A., E.G.B. and F.K.M.; formal analysis, W.A.A.; investigation, W.A.A.; resources, E.G.B. and F.K.M.; data curation, W.A.A., E.G.B. and F.K.M.; writing—original draft preparation, W.A.A.; writing—review and editing, W.A.A., E.G.B. and F.K.M.; visualization, W.A.A.; supervision, E.G.B. and F.K.M.; project administration, E.G.B. and F.K.M.; funding acquisition, E.G.B. and F.K.M. All authors have read and agreed to the published version of the manuscript.

**Funding:** The authors acknowledge funding from the Bill and Melinda Gates Foundation (BMGF) under grant number INV-005431 in support of the Excellence in Agronomy initiative. We also thank all funders who support the Sustainable Intensification of Mixed Farming Systems (MFS) and Ukama Ustawi (UU) initiatives through their contributions to the CGIAR Trust Fund.

**Data Availability Statement:** The data supporting the findings of this study are available from the corresponding author upon reasonable request.

**Conflicts of Interest:** The authors declare no conflicts of interest.

## Abbreviations

The following abbreviations are used in this manuscript:

| | |
|---|---|
| ECMWFv5.1 | European Centre for Medium-Range Weather Forecasts Version 5.1 |
| CFSv2 | NOAA Climate Prediction System Version 2 |
| CHIRPS | Climate Hazards Center InfraRed Precipitation with Stations |
| KGE′ | Modified Kling-Gupta efficiency |
| LT | Lead time |

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
