# Peer review of "Evaluating Seasonal Rainfall Forecast Gridded Models over Sub-Saharan Africa"

_hydrology, doi:10.3390/hydrology12100251_

Round 1
Reviewer 1 Report
Comments and Suggestions for Authors
The aim of this work is to assess the performance of the CFS and ECMWF-S5 models over Africa in order to evaluate their skill and to compare their reliability. The evaluation of two models and their relative performance is a key aspect of this work and is certainly appreciable. Despite this, the manuscript still presents some important drawbacks that should be addressed:
• The description (especially the acronyms) of the adopted models is not always clear. Sometimes NCEP and CFS are mentioned together. The authors should clarify precisely which models are considered in the analysis. A comparative table would also be very helpful, summarizing the main characteristics of NCEP and ECMWF-S5 (e.g., integration time, lead time, number of ensemble members, etc.).
•The subdivision adopted along line 187 is not sufficiently explained. The authors should clarify the rationale for this choice.
• The use of KGE for precipitation is unusual. The authors should explain why this index was chosen instead of more standard ones such as RMSE. Furthermore, for probabilistic forecasts, CRPS (Continuous Ranked Probability Score) would have been more appropriate.
• Please support the sentence “These models are associated with some uncertainties thus, it is necessary to determine their performance before application, typically done by comparing the forecasts to the actual precipitation.” (lines 67–69) with the reference: “Performance assessment of neural network models for seasonal weather forecast post processing in the Alpine region.”
• The sentence “Several studies have evaluated the performance of climate models over various geographical locations” should be corrected to: “Several studies have evaluated the performance of seasonal forecasting over various geographical locations.” This revised version should be supported with the reference: and “An interpretable machine learning model for seasonal precipitation forecasting.”
• In Figure 8, panel b, the trend line appears almost flat. The authors should check this result carefully and provide an explanation of why it occurs.

Reviewer 2 Report
Comments and Suggestions for Authors
The study investigates the skill of two seasonal rainfall forecasting systems, ECMWF-S5 and CFS, using records from 52 rain gauges located across Sub-Saharan Africa between 2012 and 2022. Performance is evaluated through a mix of continuous statistics such as correlation, bias and RMSE, as well as categorical scores including POD, FAR, CSI and FBI. A modified KGE index is also applied. The analysis is broken down by forecast lead time (1–3 months), temporal scale (daily and dekadal), geographic sub-regions, and elevation classes. Results indicate that ECMWF-S5 generally provides more reliable forecasts than CFS, with particularly strong performance in West Africa and at shorter leads. However, its skill weakens at higher elevations and longer lead times. CFS tends to produce overly wet forecasts, while ECMWF-S5 leans toward slight underestimation. These patterns highlight the practical value of the models for agricultural planning, but they also point to the necessity of applying bias correction. By testing forecast skill under severe data limitations, the work addresses an important gap in African hydrology. The analysis is thorough and the figures are clear, though several shortcomings need attention. In its current form, publication could be considered once the issues are addressed.
1. One major limitation is the sparse and uneven gauge network: 29 stations are in West Africa, 19 in the South, but only 4 in the East. This imbalance makes it difficult to capture regional variability, especially in the complex terrain of East Africa, and may distort continent-wide conclusions. A more detailed discussion of this issue is needed.
2. The study also mentions correcting daily rainfall series for bias, but does not describe how this was done. Without stating whether methods such as quantile mapping or linear scaling were used, the reproducibility of the work is compromised. The lack of clarity also raises uncertainty about how the correction shaped the results, which is particularly problematic when dealing with data-scarce regions.
3. Another concern is the direct comparison of individual gauge observations with model grid outputs, without any spatial interpolation or scaling. The reasoning behind this choice should be explained, together with its limitations.
4. Finally, the number of stations included in the analysis is reduced from 176 to 52. Yet the exact quality-control criteria—for example, thresholds for missing values—are not specified. Providing this information would improve transparency and strengthen the credibility of the results.
Round 2
Reviewer 1 Report
Comments and Suggestions for Authors
The paper is now worth for publication.